# Prevalence of Elevated Blood Triglycerides and Associated Risk Factors: Findings from a Nationwide Health Screening in Mongolia

**DOI:** 10.3390/ijerph21121559

**Published:** 2024-11-25

**Authors:** Khangai Enkhtugs, Tumur-Ochir Tsedev-Ochir, Enkhtur Yadamsuren, Batzorig Bayartsogt, Bayarbold Dangaa, Otgonbat Altangerel, Oyuntugs Byambasukh, Oyunsuren Enebish

**Affiliations:** 1Department of Family Medicine, School of Medicine, Mongolian National University of Medical Sciences, Ulaanbaatar 14210, Mongolia; khangai@mnums.edu.mn; 2State Central Third Hospital, Ulaanbaatar 16081, Mongolia; tsch@shastinhospital.mn; 3Deprtment of Dermatology, School of Medicine, Mongolian National University of Medical Sciences, Ulaanbaatar 14210, Mongolia; enkhtur@mnums.edu.mn; 4Department of Epidemiology and Biostatistics, School of Public Health, Mongolian National University of Medical Sciences, Ulaanbaatar 14210, Mongolia; batzorig@mnums.edu.mn (B.B.); bayarbold@moh.gov.mn (B.D.); 5Ministry of Health, Ulaanbaatar 14253, Mongolia; 6Department of Hematology, School of Medicine, Mongolian National University of Medical Sciences, Ulaanbaatar 14210, Mongolia; otgonbat.a@mnums.edu.mn; 7Department of Endocrinology, School of Medicine, Mongolian National University of Medical Sciences, Ulaanbaatar 14210, Mongolia

**Keywords:** lipids, cardiovascular health, obesity

## Abstract

Background: This study aims to assess the demographic, lifestyle, and clinical characteristics associated with varying levels of triglycerides (TGs) in a large population sample. Methods: This cross-sectional study utilized data from a nationwide health screening program in Mongolia. A total of 125,330 participants (mean age: 43.8 ± 15.3 years) were included. TG levels were categorized into normal, borderline high, high, and very high. Due to the small number of participants in the very high TG group, they were combined with the high TG category for analysis. Multivariate logistic regression was performed to identify independent predictors of elevated TG levels. Results: The majority of participants (80.3%) had normal TG levels, while 10.3% had borderline high, 8.7% had high, and 0.7% had very high TG levels. Significant predictors of elevated TG levels included age (OR 1.013, 95% CI 1.012–1.014), male (OR 2.328, 95% CI 2.251–2.408), obesity (OR 1.920, 95% CI 1.855–1.987), central obesity (OR 1.866, 95% CI 1.801–1.933), smoking (OR 1.399, 95% CI 1.347–1.453), alcohol use (OR 1.233, 95% CI 1.176–1.292), and non-regular exercise (OR 1.144, 95% CI 1.118–1.171). Sex-specific analysis revealed that elevated TG levels were more prevalent among males, regardless of other risk factors such as obesity and smoking. Conclusions: Male sex, obesity, and smoking were the strongest predictors of elevated TG levels.

## 1. Introduction

Elevated triglyceride (TG) levels are an increasingly recognized risk factor for cardiovascular disease (CVD) worldwide. As a primary component of blood lipids, triglycerides play a crucial role in energy storage and metabolism. However, when TG levels are abnormally high, they contribute to the development of atherosclerosis, metabolic syndrome, and other cardiovascular complications [1]. Hypertriglyceridemia has gained attention as a significant public health issue, particularly in developing and transitional economies where rapid urbanization and lifestyle changes are prevalent [2]. The burden of elevated TG levels is closely linked to modifiable risk factors such as obesity, poor dietary habits, physical inactivity, and substance use, all of which are on the rise in these regions [3,4].

Mongolia, like many other countries in Asia, is experiencing rapid socioeconomic changes that are accompanied by shifts in lifestyle, diet, and health behaviors. Despite this, limited data exist on the prevalence of elevated TG levels and their associated risk factors in the Mongolian population. Previous studies in other populations have consistently found that male sex, central obesity, smoking, and alcohol consumption are strongly associated with hypertriglyceridemia [5,6]. Understanding the epidemiology of TG levels and the demographic and clinical characteristics that contribute to elevated levels in Mongolia is essential for designing effective prevention and intervention strategies.

This study aims to address this gap by investigating the prevalence of elevated TG levels among participants of a nationwide health screening program in Mongolia. The study also examines the demographic, clinical, and lifestyle factors that are most closely associated with high TG levels, providing insights into the unique challenges faced in managing lipid disorders in this population.

## 2. Materials and Methods

### 2.1. Study Participants

This study utilized data from a nationwide health screening program conducted by the Ministry of Health in Mongolia between 2022 and 2023. The initial dataset included 209,055 adults. Participants with missing data, outliers related to key variables, and those who were non-fasting were excluded, resulting in a final sample of 125,330 participants for analysis.

The study was conducted in compliance with the principles of the Helsinki Declaration and was approved by the Medical Ethics Committee of the Ministry of Health, Mongolia (Approval No: 23/042, dated 5 July 2023).

### 2.2. Data Collection and Study Variables

Details for data collection have been described elsewhere [7]. Participants were instructed to fast overnight (for at least 8 h) before blood samples were drawn the following morning. Fasting status was verified through both participant responses and diagnostic records. The collected data included demographic factors (age, sex, education level, and living area), clinical measures (BMI, waist circumference, and blood pressure), and lifestyle factors (smoking status, alcohol use, dietary habits, and physical activity).

Demographic information was recorded during the health screenings, including age, sex, education level, living area (urban vs. rural), smoking status, alcohol use, fruit and vegetable intake, and physical activity. Education level was categorized as “lower” (≤4 years of education) or “above”. Marital status was classified as “married/cohabitant” or “others”, and living area was divided into urban and rural. Smoking status was defined as current smoker (including those who quit within the last 6 months) or never smoked. Alcohol use was assessed based on consumption within the last 30 days, with different thresholds for men and women. Fruit and vegetable intake was categorized according to WHO STEPS criteria, where consuming 5 or more servings per week was considered sufficient. Physical activity was defined as regularly engaging in activities equivalent to 10,000 steps daily.

Standardized protocols were followed for measuring body weight, height, waist circumference, and blood pressure. BMI was calculated using height and weight, and categorized as normal weight, overweight (25–29.9 kg/m^2^), or obese (≥30 kg/m^2^). Central obesity was defined as a waist circumference of ≥90 cm for men and ≥80 cm for women.

Triglyceride levels were measured using a standardized enzymatic method in blood plasma across all screening centers. Based on international guidelines, particularly those from the National Cholesterol Education Program (NCEP) Adult Treatment Panel III (ATP III) and the American Heart Association (AHA), participants were categorized into the following groups based on triglyceride levels: normal (<1.7 mmol/L), borderline high (1.7–2.25 mmol/L), high (2.25–5.64 mmol/L), and very high (>5.65 mmol/L) [1,2]. Due to the limited number of participants in the very high category, they were combined with the high category for further analysis.

### 2.3. Statistical Analysis

The characteristics of the study population were expressed as means with standard deviations (SDs) for continuous variables and as percentages for categorical variables, stratified by triglyceride categories. Tukey’s five-number summary (minimum, first quartile, median, third quartile, maximum) was calculated for continuous variables and presented in Appendix A, complementing the traditional mean and standard deviation. Differences between groups were analyzed using ANOVA for continuous variables and Pearson’s chi-square test for categorical variables. The normality of continuous variables was assessed using the Shapiro–Wilk test. All variables were found to follow a normal distribution, and thus, parametric tests were applied in the analysis.

In the logistic regression analysis, elevated triglyceride levels (borderline high, high, and very high combined) were used as the dependent variable, with normal triglyceride levels serving as the reference group. To assess potential multicollinearity among the predictors, we calculated the Variance Inflation Factor (VIF) for each variable. All VIF values were below 1.5, indicating that multicollinearity was negligible and did not impact the stability or reliability of the regression estimates. The results are presented as odds ratios (ORs) with corresponding 95% confidence intervals (CIs). In addition to categorizing triglyceride (TG) levels as elevated versus normal, we also conducted a multivariate linear regression analysis treating TG levels as a continuous variable. This approach allowed us to capture the full variability in TG levels and provided a quantitative comparison of the effects of the predictors. The results from this continuous analysis, including unstandardized beta coefficients, are presented below to complement the discrete analysis and offer a more comprehensive assessment.

Additional analyses were conducted by stratifying participants into subgroups based on sex, age, obesity, and central obesity to explore more detailed associations. We also explored potential interaction effects between sex and key predictors, including obesity and smoking, to evaluate whether the association between these factors and triglyceride levels varied by sex. Interaction terms for sex x obesity and sex x smoking were included in the multivariate regression models. The results of these interactions were examined to determine if the impact of obesity and smoking on triglyceride levels differed between males and females.

Statistical analyses were performed using IBM SPSS V.28.0, and graphical representations were generated using GraphPad Prism 9.0. A *p*-value of <0.05 was considered statistically significant for all analyses. A 95% confidence interval was used to determine statistical significance rather than focusing solely on *p*-values.

## 3. Results

The study included 125,330 participants with a mean age of 43.8 ± 15.3 years. The majority of participants were female (60.6%). Triglyceride levels were categorized into four groups, with the majority (80.3%) of participants having normal triglyceride levels. Meanwhile, 10.3% fell within the borderline high range, 8.7% had high triglyceride levels, and only 0.7% had very high triglyceride levels. Due to the small numbers in the high and very high groups, these categories were combined for further analysis. The characteristics of the study population across the three triglyceride categories are shown in Table 1.

Significant differences in age were observed across triglyceride (TG) categories (*p* < 0.001). Participants with borderline high TG levels (1.7–2.25 mmol/L) had a mean age of 47.2 ± 14.5 years, while those with high TG levels (above 2.25 mmol/L) had a mean age of 46.9 ± 13.7 years, compared to 43.0 ± 15.5 years in the normal TG group. A higher proportion of males were present in the elevated TG categories. The percentage of males increased from 35.4% in the normal TG category to 49.7% in the borderline high category and 61.1% in the high TG category (*p* < 0.001). Regarding living area, participants residing in urban areas had higher TG levels, with 47.1% of those in the borderline high category and 45.1% in the high category living in urban areas (*p* < 0.001). The percentage of participants with lower education levels was slightly reduced in the borderline high TG category (7.8%) compared to the normal TG category (8.8%) (*p* < 0.001).

BMI was significantly higher in participants with elevated TG levels, reaching 28.5 ± 4.9 kg/m^2^ in the borderline high category and 29.1 ± 4.7 kg/m^2^ in the high category, compared to 26.0 ± 4.7 kg/m^2^ in the normal category (*p* < 0.001). Obesity was more prevalent in participants with elevated TG levels, with 34.6% of those in the borderline high category and 39.3% in the high category classified as obese (*p* < 0.001). Waist circumference measurements followed a similar trend, with higher circumferences observed in participants with elevated TG levels. Among males, waist circumference was 95.5 ± 14.3 cm in the high TG category, compared to 86.5 ± 13.5 cm in the normal category (*p* < 0.001). Among females, waist circumference was 91.2 ± 13.9 cm in the high TG category compared to 83.5 ± 13.4 cm in the normal category (*p* < 0.001). The prevalence of central obesity was higher in participants with elevated TG levels. Central obesity was present in 69.2% of participants in the borderline high category and 70.9% in the high category, compared to 51.3% in the normal category (*p* < 0.001).

Blood pressure was also significantly higher in participants with elevated TG levels. Systolic blood pressure was 125.0 ± 16.5 mmHg in the high TG category compared to 118.9 ± 15.6 mmHg in the normal category (*p* < 0.001). Diastolic blood pressure followed a similar pattern, with readings of 81.0 ± 10.9 mmHg in the high TG category compared to 76.7 ± 10.4 mmHg in the normal category (*p* < 0.001). Total cholesterol levels were highest in the borderline high TG category (5.54 ± 1.21 mmol/L) and the high TG category (5.51 ± 1.79 mmol/L), compared to 4.95 ± 1.14 mmol/L in the normal category (*p* < 0.001).

Smoking and alcohol use were more prevalent in participants with elevated TG levels. Smoking rates increased from 17.1% in the normal category to 26.0% in the borderline high category and 32.3% in the high category (*p* < 0.001). Similarly, alcohol use increased from 8.2% in the normal category to 11.6% in the borderline high category and 15.2% in the high category (*p* < 0.001). In terms of dietary habits, daily fruit and vegetable consumption was slightly lower in participants with elevated TG levels (*p* < 0.001). The percentage of participants engaging in regular exercise and physical activity was also slightly lower in the high TG category (59.0%) compared to the normal category (60.0%) (*p* < 0.001).

As shown in Table 2, univariate analysis revealed that age, male, obesity, central obesity, smoking, alcohol use, and inadequate fruit and vegetable consumption were all significantly associated with an increased risk of elevated TG levels (*p* < 0.001 for all). In the multivariate model, age (OR 1.013, 95% CI 1.012–1.014), male (OR 2.328, 95% CI 2.251–2.408), obesity (OR 1.920, 95% CI 1.855–1.987), central obesity (OR 1.866, 95% CI 1.801–1.933), smoking (OR 1.399, 95% CI 1.347–1.453), alcohol use (OR 1.233, 95% CI 1.176–1.292), and non-regular exercise (OR 1.144, 95% CI 1.118–1.171) remained significant predictors of elevated TG levels. Inadequate fruit and vegetable consumption did not reach statistical significance in the multivariate analysis (OR 1.009, 95% CI 0.991–1.027, *p* = 0.312).

In addition to analyzing triglyceride levels as a categorical variable (elevated vs. normal), we also performed a multivariate analysis treating TG levels as a continuous variable to provide a more detailed understanding of the relationships between predictors and TG levels. The results of this continuous analysis, including the unstandardized beta coefficients, are presented in Table 3, offering deeper insights into the quantitative impact of predictors on triglyceride levels.

In the regression analysis, sex, obesity status, and smoking showed stronger effects compared to other variables. Therefore, sex differences were first analyzed, showing that males had a higher prevalence of elevated TG levels (Figure 1). Specifically, 13.3% of males had high TG levels compared to only 5.7% of females. Additionally, 1.3% of males had very high TG levels, while this was only 0.3% in females.

Although these factors are independent, further analysis was conducted to examine how sex plays a role in influencing TG levels. Figure 2 and Figure 3 suggest that males have a higher prevalence of elevated TG levels, but smoking status also plays a significant role in both males and females. For instance, among female smokers, 11.3% had high TG levels, compared to only 5.8% among non-smokers. In males, 16.2% of smokers had high TG levels compared to 13.4% of non-smokers.

Additionally, obesity plays a significant role in influencing TG levels in both males and females. Among obese males, 27.2% had high TG levels compared to only 6.5% among normal-weight males. Similarly, among obese females, 10.5% had high TG levels, compared to just 2.9% in normal-weight females (Figure 3).

In addition to the main effects, we explored potential interaction effects between sex and key predictors, specifically obesity and smoking. The analysis showed significant interactions between sex and both obesity and smoking. The interaction between sex and obesity revealed that the impact of obesity on elevated triglycerides was stronger in males (OR = 3.03, 95% CI: 2.90–3.17) compared to females (OR = 2.47, 95% CI: 2.37–2.58, *p* < 0.001). Similarly, the sex and smoking interaction indicated that smoking had a more pronounced effect on triglyceride levels in females (OR = 1.96, 95% CI: 1.80–2.12) than in males (OR = 1.25, 95% CI: 1.20–1.30, *p* < 0.001). These findings suggest that the risk associated with these factors varies by sex, highlighting the importance of considering gender-specific interventions for triglyceride management.

## 4. Discussion

The findings from this study highlight a significant burden of elevated triglyceride levels in Mongolia, with nearly 20% of the study population exhibiting borderline high, high, or very high TG levels. The results of this study underscore the importance of addressing modifiable risk factors such as obesity, smoking, and alcohol consumption to mitigate the rising incidence of hypertriglyceridemia and its associated cardiovascular risks.

The prevalence of elevated triglyceride levels in this Mongolian population is consistent with patterns observed in other Asian countries, such as China and South Korea, where males also exhibit higher TG levels, and central obesity strongly correlates with elevated TGs [8,9]. This reflects our findings that central obesity is a key determinant of TG levels in Mongolia. Interestingly, in Japan, TG levels tend to be lower compared to other Asian nations, possibly due to dietary habits that include higher fish consumption and lower saturated fat intake [10]. When compared to Western countries, such as those in Europe and the United States, the prevalence of elevated TG levels in Mongolia appears somewhat higher. For instance, in the United States, around 25% of adults have hypertriglyceridemia (TG levels ≥150 mg/dL), with males exhibiting higher rates [11]. This suggests that while TG levels are rising globally, regional dietary patterns and lifestyle factors may contribute to the differences observed between Mongolia and other countries.

One of the most striking findings is the strong association between male sex and elevated TG levels. The study found that males had significantly higher TG levels across all categories, even after accounting for other factors such as smoking, alcohol use, and obesity. This is consistent with previous research, which indicates that males are more susceptible to lipid abnormalities due to hormonal differences, lower estrogen levels, and a tendency to accumulate more visceral fat, which is metabolically active and linked to increased TG production [12,13]. Furthermore, evidence suggests that the association between elevated TG levels and cardiovascular risk may differ between men and women. Although men tend to have higher absolute TG levels, research shows that women, particularly postmenopausal women, may experience a greater relative increase in cardiovascular risk at comparable TG levels due to hormonal changes and the loss of estrogen’s protective effects [4,14]. Studies indicate that postmenopausal women have a higher cardiovascular risk associated with elevated TGs compared to premenopausal women and may even have a similar or higher risk than men at similar TG levels [14,15]. This highlights the need for sex-specific interventions in triglyceride management for both men and women, taking into account their unique cardiovascular risk profiles.

Obesity, especially central obesity, emerged as a critical determinant of elevated TG levels. The study showed a clear gradient in waist circumference and body mass index (BMI) across TG categories, with significantly higher measurements in those with borderline high and elevated TG levels. Central obesity is a well-known risk factor for metabolic syndrome and is closely tied to insulin resistance, which drives the accumulation of TGs in the bloodstream [16,17]. Given the high prevalence of obesity observed in this study, there is an urgent need for interventions that promote healthy weight management through lifestyle changes such as improved diet and regular physical activity [18].

Smoking and alcohol consumption also demonstrated a strong association with elevated TG levels. The harmful effects of smoking on lipid metabolism have been well documented, with studies showing that smoking increases the hepatic production of very-low-density lipoprotein (VLDL), leading to higher TG levels [19]. Additionally, the oxidative stress induced by smoking can impair lipid clearance, exacerbating hypertriglyceridemia [20]. The relationship between alcohol consumption and TG levels is similarly well established, as alcohol promotes hepatic lipogenesis while inhibiting lipoprotein lipase, resulting in reduced TG clearance [21]. The findings from this study reinforce the need for public health campaigns aimed at reducing smoking and excessive alcohol use as part of a comprehensive strategy to manage dyslipidemia [22].

While dietary factors like fruit and vegetable consumption did not remain significant in the multivariate analysis, the role of diet in lipid management should not be overlooked. Diets high in refined carbohydrates and unhealthy fats are known contributors to elevated TG levels [23]. Promoting a balanced diet rich in fruits, vegetables, and healthy fats remains a cornerstone of cardiovascular risk reduction strategies [24].

The study also highlights the importance of physical activity in maintaining healthy TG levels. Although the difference in physical activity levels between those with normal and elevated TG levels was modest, regular exercise is known to enhance TG metabolism by increasing lipoprotein lipase activity and improving insulin sensitivity [25]. The slightly lower levels of physical activity observed in participants with high TG levels suggest that encouraging consistent physical activity could be a key intervention point in this population [26].

Urban–rural disparities in TG levels were also noted, with higher TG levels observed among urban residents. This likely reflects the impact of urbanization, where access to processed foods, sedentary lifestyles, and higher stress levels contribute to dyslipidemia [27]. Similar findings have been reported in other countries experiencing rapid urbanization, where the shift toward unhealthy lifestyle practices has led to an increase in lipid disorders [28]. Addressing these disparities requires targeted public health efforts that account for the unique challenges faced by urban populations in Mongolia.

The limitations of this study include its cross-sectional design, which limits causal inferences, and the potential for reporting bias in self-reported lifestyle factors. Despite these limitations, the study’s large sample size and comprehensive data collection enhance the reliability of the findings. Future research should consider longitudinal designs to explore how changes in lifestyle and clinical factors influence TG levels over time. Additionally, integrating traditional Mongolian medicine with modern approaches could offer culturally relevant solutions for managing lipid disorders [29,30].

## 5. Conclusions

In conclusion, this study provides important insights into the prevalence and determinants of elevated triglyceride levels in Mongolia. Male sex, central obesity, smoking, and alcohol use are key contributors to hypertriglyceridemia in this population. Public health strategies should prioritize addressing these modifiable risk factors to reduce cardiovascular risk. Tailored interventions focusing on high-risk groups, including men and those with obesity, are essential for managing the growing burden of dyslipidemia in Mongolia.

## Figures and Tables

**Figure 1 ijerph-21-01559-f001:**
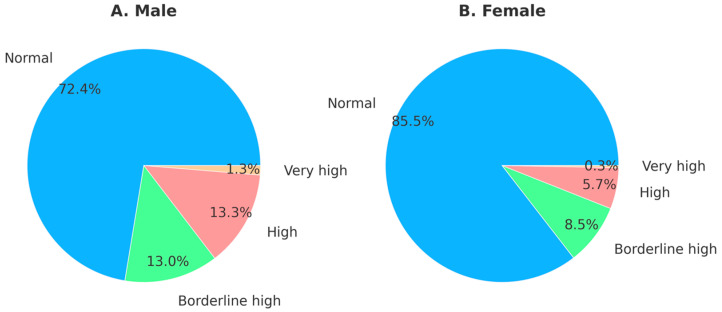
Sex differences in triglyceride levels.

**Figure 2 ijerph-21-01559-f002:**
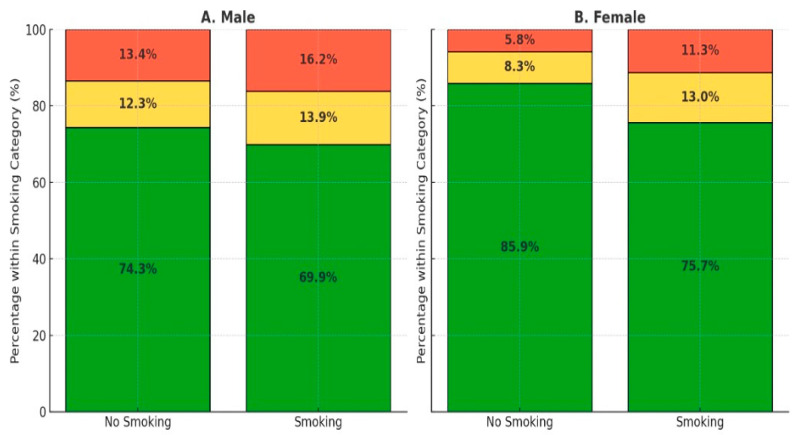
Smoking status and triglyceride levels by sex.

**Figure 3 ijerph-21-01559-f003:**
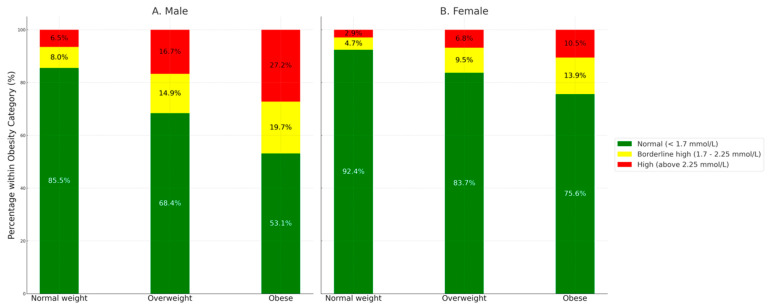
Obesity status and triglyceride levels by sex.

**Table 1 ijerph-21-01559-t001:** Characteristics of study population.

Findings	Total	Triglyceride Category			
		Normal	Borderline High	High	*p*-Value
Age (years)	43.8 ± 15.3	43.0 ± 15.5	47.2 ± 14.5	46.9 ± 13.7	<0.001
Male, % (*n*)	39.4 (49,270)	35.4 (35,676)	49.7 (6412)	61.1 (7182)	<0.001
Living area: Urban, % (*n*)	41.7 (52,280)	40.6 (40,905)	47.1 (6076)	45.1 (5299)	<0.001
Education: lower, % (*n*)	8.6 (10,782)	8.8 (8821)	7.8 (1001)	8.2 (960)	<0.001
Marital status: Married or cohabitant, % (*n*)	79.4 (99,563)	78.0 (78,525)	84.3 (10,783)	86.5 (10,165)	<0.001
BMI (kg/m^2^)	26.6 ± 4.9	26.0 ± 4.7	28.5 ± 4.9	29.1 ± 4.7	<0.001
Obesity, % (*n*)	22.3 (27,901)	18.7 (18,813)	34.6 (4468)	39.3 (4620)	<0.001
Waist Circumference (male, cm)	88.7 ± 14.2	86.5 ± 13.5	93.5 ± 14.3	95.5 ± 14.3	<0.001
Waist Circumference (female, cm)	84.5 ± 13.7	83.5 ± 13.4	90.6 ± 13.9	91.2 ± 13.9	<0.001
Central obesity (male, %, *n*)	45.4 (22,358)	38.4 (13,685)	60.7 (3893)	66.6 (4780)	<0.001
Central obesity (female, %, *n*)	61.1 (46,506)	58.3 (37,918)	77.6 (5041)	77.8 (3547)	<0.001
Systolic (mmHg)	119.9 ± 15.9	118.9 ± 15.6	124.3 ± 16.6	125.0 ± 16.5	<0.001
Diastolic (mmHg)	77.4 ± 10.6	76.7 ± 10.4	80.1 ± 10.8	81.0 ± 10.9	<0.001
Total cholesterol (mmol/L)	5.06 ± 1.24	4.95 ± 1.14	5.54 ± 1.21	5.51 ± 1.79	<0.001
Smoking, % (*n*)	19.5 (24,383)	17.1 (17,230)	26.0 (3361)	32.3 (3792)	<0.001
Alcohol use, % (*n*)	9.2 (11,537)	8.2 (8250)	11.6 (1503)	15.2 (1784)	<0.001
Fruit and vegetable daily use, % (*n*)	25.6 (32,062)	25.8 (25,940)	25.4 (3278)	24.2 (2844)	<0.001
Regular exercise and PA, % (*n*)	60.2 (75,396)	60.0 (60,405)	62.4 (8056)	59.0 (6935)	<0.001

Data are presented as mean ± SD and percentages (numbers). The *p*-values correspond to ANOVA for continuous variables and Pearson’s chi-square test for categorical variables. The high TG category includes both high (2.26–5.64 mmol/L) and very high (≥5.65 mmol/L) triglyceride levels.

**Table 2 ijerph-21-01559-t002:** Logistic regression analysis for association with increased TG risk.

Variables	Univariable OR	95% CI		*p* Value	Multivariable OR	95% CI		*p* Value
		Lower Bound	Upper Bound			Lower Bound	Upper Bound	
Age (years)	1.02	1.02	1.02	<0.01	1.01	1.01	1.01	<0.01
Sex: Male	2.24	2.18	2.31	<0.01	2.33	2.25	2.41	<0.01
Obesity	2.54	2.47	2.62	<0.01	1.92	1.86	1.99	<0.01
Central Obesity	2.22	2.16	2.29	<0.01	1.87	1.80	1.93	<0.01
Smoking	1.98	1.92	2.05	<0.01	1.40	1.35	1.45	<0.01
Alcohol use	1.72	1.65	1.80	<0.01	1.23	1.18	1.29	<0.01
Inadequate fruit and vegetable use	1.04	1.02	1.06	<0.01	1.01	0.99	1.03	0.31
Non-regular exercise and physical activity	1.14	1.12	1.17	<0.01	1.14	1.12	1.17	<0.01

Data presented as odds ratio with 95% confidence intervals (95% CI). OR, odds ratio, CI, confidence intervals. The high TG category includes both high (2.26–5.64 mmol/L) and very high (≥5.65 mmol/L) triglyceride levels.

**Table 3 ijerph-21-01559-t003:** Continuous regression analysis for TG levels.

Variables	Unstandardized β Coefficient * (mmol/L)	95% CI		*p* Value
		Lower Bound	Upper Bound	
Age (years)	0.01	0.01	0.01	<0.01
Sex: Male	0.35	0.34	0.36	<0.01
Obesity	0.28	0.27	0.29	<0.01
Central Obesity	0.20	0.19	0.21	<0.01
Smoking	0.13	0.11	0.14	<0.01
Alcohol use	0.10	0.08	0.11	<0.01
Inadequate fruit and vegetable use	0.01	0.01	0.02	<0.01
Non-regular exercise and physical activity	0.07	0.07	0.08	<0.01

Data presented as unstandardized β coefficients with 95% confidence intervals (95% CI). * Multivariate analysis.

## Data Availability

The data used to support the findings of this study are available from the corresponding author upon request.

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
