# Peer review of "Prevalence of Elevated Blood Triglycerides and Associated Risk Factors: Findings from a Nationwide Health Screening in Mongolia"

_ijerph, 2024, doi:10.3390/ijerph21121559_

Round 1
Reviewer 1 Report
Comments and Suggestions for Authors
In the manuscript “Prevalence of elevated triglycerides and associated risk factors: Findings from a nationwide health screening in Mongolia”, Enkhtugs et al. analyze the relationship between the levels of TGs and different socioeconomical and medical parameters, both by univariate and multivariate analyses, in a cohort in Mongolia. The study is interesting and the manuscript is focused and well written. The following concerns should be considered to accept its publication.
Major points
1. Blood analysis. In materials and methods it is not clear if the characterization of TGs was done in blood serum or blood plasma. It should be specified.
2. Summarizing data. The authors report the different parameters as the mean plus/minus the standard deviation. However, this reporting does not allow evaluating if the distribution of the parameter is skewed. Consequently, the different values in Table 1 should be complemented with Tukey’s five-numbers in order to obtain a better description of the characteristics of the population and its relationship with the levels of TGs.
3. p-values and complete reporting.
a. Despite the tradition of reporting p-values as inequalities, nowadays all statisticians require to report p-values as numerical values [https://doi.org/10.1080/00031305.2016.1154108, https://doi.org/10.1080/00031305.2019.1583913].
b. The same references require that all p-values should be reported (complete reporting), even if they are not “statistically significant” (e.g. in Table 2).
c. Considering the statistical power of having more than 2E5 samples, any hypothesis test on the mean or median will be statistically significant, even if the effect is minor. This is especially important in logistic regression considering OR, where the quantitative effect of the predictor is absent. Therefore, the limit of statistical significance at p < 0.05 is rather useless. In fact, most predictors are “statistically significant” in Table 2. Rather than setting statistical significance p < 0.05, the authors should state that the level of confidence for the confidence intervals was 0.95.
4. Description of the dataset (Table 1).
a. p-values are reported in the last column. However, it is not clear to which hypothesis test it corresponds to. Is it to ANOVA, Kruskal-Wallis or another type of tests for the groups Normal, Borderline High, and High? This should specified in the caption of Table 1. As commented before, p-values should be reported in a numerical way.
b. Waist circumference has different criteria for men and women and it is correctly stratified in Table 1. Central obesity also has different criteria for men and women according to section 2.2. However, central obesity is not stratified for sex in Table 1.
c. Marital status is mentioned section 2.2 as predictor. However, it is not reported in Table 1. In addition, was it included in the univariate and multivariate regression?
5. Correlation and multicollinearity analysis of the predictors is necessary for multivariate regression.
a. High correlation among predictors (multicollinearity) can distort the estimation of regression coefficients and ORs, making it difficult to determine the individual effect of each variable. By performing correlation analysis, researchers can identify and address multicollinearity, ensuring the model stability and reliability (multicollinearity overinflates the standard errors). In fact, the inclusion of correlating predictors can spoil the regression analysis [DOI: 10.1088/1742-6596/949/1/012009].
b. Correlation could be, for example, the cause of the minor effect of the predictor “Inadequate fruit and vegetable use” in the multivariate analysis (Table 2). As the authors discussed, it is striking the minor effect of this predictor. The cause could be that it is correlating with other predictors. As a coefficient in multivariate regression represent the effect once the other predictors have been taken into account, the low OR of “Inadequate fruit and vegetable use” might be caused by an unexplored correlation with other predictors. Hence, exploring the correlation among the predictors will improve the data interpretation and the discussion of the manuscript.
6. Discretization of TG data. Forcing continuous response variables into discrete variables hampers the statistical analysis and can lead to loss of information and reduced statistical power. When TGs are forced into discrete categories, the variability within each category is ignored, potentially oversimplifying relationships. Furthermore, continuous TG data provide more detailed insights in the data and allow for quantitative comparison of the predictors. Consequently, the current analysis with discretized data (elevated TGs vs normal) providing odds ration should be complemented with a multivariate modeling having TGs as a continuous response variable. The coefficients of such analysis can be added to Table 2.
7. Table 2. Only “statistically significant” predictors are reported. Nevertheless, as said in point 3, statisticians require complete reporting. Thus the OR, their confidence intervals, and their p-values (in a numerical way) should be reported for all the predictors.
8. Sex/Gender.
a. Distinguishing sex and gender. Sex refers to biological differences between males and females, such as chromosomes, hormones, and reproductive organs. Gender, on the other hand, is a social and cultural concept related to roles, behaviors, and identities [ https://kjonnsforskning.no/sites/default/files/what_is_the_gender_dimension_roggkorsvik_kilden_genderresearch.no_.pdf, page 8]. It should be clarified if the predictor gender in fact refers to sex. This is of crucial importance, as testosterone and estrogens are fundamental to determine the level of lipogenesis in the liver and the body (hence TGs in plasma/serum). The effect of sexual hormones is associated with sex but not necessarily with gender (although there is a clear correlation between sex and gender). As the discussion about gender refers to sexual hormones (line 214), I think that the authors really refer to sex, rather than to gender.
b. After spotting gender in the multivariate regression, the authors analyze the effect of gender/sex back on raw data, evaluating the different levels of smoking and obesity according to gender/sex. However, in this type of analysis the confounding effects are difficult to establish and prone to Simpson’s paradox [DOI: 10.1093/ije/dyr041]. A proper analysis of the differences between man and women should consider the regression coefficients when the confounding factors (smoking, obesity, etc) are considered, including interaction factors between gender/sex and smoking or obesity. As the coefficients of the regression analysis evaluate the influence of a factor when the influence of the other factors has been considered, the comparison of models with and without interaction allow to study the true association of TGs with male/female when the other factors have been accounted for.
c. In the discussion, it is assumed that the same levels of TGs predict the same cardiovascular risk for men and women suggesting a special focus to intervene males in Mongolia to reduce their cardiovascular risk. However, it is not clear that the assumption is correct. To really support this conclusion, the authors should check in the bibliography and discuss if, all other equal, the same levels of TGs predict the same risk of cardiovascular risk for men and women.
9. Discussion with the results of similar datasets in other countries. For example, are there differences with Western or other Asian countries?
Minor points
1. The title should specify that it is TGs in blood
2. Table 1. In the last row, one guesses that PA refers to physical activity. This should be specified in the caption.
3. Table 2. The caption should specify that CI corresponds to “confidence interval” and OR to “odds ratio”
4. In the abstract it is reported that the authors used logistic regression. However, it is not clearly stated in the methodology
Conclusions
The manuscript is of interest. However, several changes should be made for improving its analysis and impact.
Author Response
Thank you very much for your very valuable comments and improved our manuscript. Enclosed is my detailed point-by-point response for your consideration.

Reviewer 2 Report
Comments and Suggestions for Authors
In the manuscript “Prevalence of elevated triglycerides and associated risk factors: 2 Findings from a nationwide health screening in Mongolia “the authors have presented some basic findings. The study is poorly designed and is only a descriptive survey. The mechanism of action, importance of this factor, novelty of work, etc., are unknown.
Comments on the Quality of English LanguageMinor editing of English language required.
Author Response
Response: While the study is descriptive, it fills an important gap by offering new insights into the epidemiological landscape of triglycerides in a country undergoing significant lifestyle changes. To address concerns regarding the mechanism of action and novelty, revisions have been made to the manuscript. The discussion has been expanded to emphasize the potential public health implications of elevated triglycerides in this population and to explore possible mechanistic pathways linking triglycerides to cardiovascular risk. The novelty of the findings has been clarified by comparing the results with those from other countries and highlighting their importance in the Mongolian context. Additionally, we have further expanded the discussion to explain how the findings can contribute to the development of targeted public health interventions in Mongolia, with a focus on key modifiable risk factors.
Round 2
Reviewer 1 Report
Comments and Suggestions for Authors
Enkhtugs et al. have improved the manuscript and its accuracy and completeness are stronger. The authors have solved most of my concerns and suggestions. However, some points remain to make it publishable.
-
Response to Major point 3a, p-values in a numerical way. Despite the authors claim to stop reporting p-values as inequalities, in Tables 1 and 2 they are still as inequalities in the version 2 of the manuscript. This point should be fixed.
-
Response to Major point 3c, statistical significance. The authors respond “…, we have specified in the manuscript that a 95% confidence interval was used rather than focusing solely on p-values for statistical significance.” However, such statement is not present in the uploaded version
-
Response to Major point 5a, multicollinearity. The authors have clarified this point in the response. However, they have not included that all VIF values were 1.5 in the manuscript. They should include the multicollinearity study they did so other readers can have a stronger trust in their results.
-
Response to Major point 6. I kindly thank the authors to follow my piece of advice. However, the presentation of the results on the continuous regression on TGs must be improved. Table 2 has become a bit too crowded. I recommend to report the continuous regression in a new table (Table 3). In addition, 1) the units of the beta coefficients should be clearly reported next to the coefficient to make it easier to the reader; and 2) as said before, p-values are still as inequalities.
-
Response to Minor point 2. The meaning of PA in Table 1, at least in the version I got, has not been clarified
Author Response
Thank you for your valuable comments, which have greatly improved our manuscript. Please find attached our responses to each of the comments.
